# The Chemical Deformation of a Thermally Cured Polyimide Film Surface into Neutral 1,2,4,5-Benzentetracarbonyliron and 4,4′-Oxydianiline to Remarkably Enhance the Chemical–Mechanical Planarization Polishing Rate

**DOI:** 10.3390/nano15060425

**Published:** 2025-03-10

**Authors:** Man-Hyup Han, Hyun-Sung Koh, Il-Haeng Heo, Myung-Hoe Kim, Pil-Su Kim, Min-Uk Jeon, Min-Ji Kim, Woo-Hyun Jin, Kyoo-Chul Cho, Jinsub Park, Jea-Gun Park

**Affiliations:** 1Department of Nanoscale Semiconductor Engineering, Hanyang University, Seoul 04763, Republic of Korea; aksguq06@hanyang.ac.kr (M.-H.H.); rhgustjd09@gmail.com (P.-S.K.); 2Department of Electronic Engineering, Hanyang University, Seoul 04763, Republic of Korea; psk6208@naver.com (H.-S.K.); heo5170@naver.com (I.-H.H.); ck05102@naver.com (M.-H.K.); mujeon1214@gmail.com (M.-U.J.); a2024157411@hanyang.ac.kr (M.-J.K.); jinwh7@naver.com (W.-H.J.); kccho12@naver.com (K.-C.C.); jinsubpark@hanyang.ac.kr (J.P.)

**Keywords:** polyimide film CMP, CMP slurry, wet ceria abrasives, substitution nucleophilic bimolecular reaction (SN2 reaction), ring-opening reaction, proton transfer reaction

## Abstract

The rapid advancement of 3D packaging technology has emerged as a key solution to overcome the scaling-down limitation of advanced memory and logic devices. Redistribution layer (RDL) fabrication, a critical process in 3D packaging, requires the use of polyimide (PI) films with thicknesses in the micrometer range. However, these polyimide films present surface topography variations in the range of hundreds of nanometers, necessitating chemical–mechanical planarization (CMP) to achieve nanometer-level surface flatness. Polyimide films, composed of copolymers of pyromellitimide and diphenyl ether, possess strong covalent bonds such as C–C, C–O, C=O, and C–N, leading to inherently low polishing rates during CMP. To address this challenge, the introduction of Fe(NO_3_)_3_ into CMP slurries has been proposed as a polishing rate accelerator. During CMP, this Fe(NO_3_)_3_ deformed the surface of a polyimide film into strongly positively charged 1,2,4,5-benzenetetracarbonyliron and weakly negatively charged 4,4′-oxydianiline (ODA). The chemically dominant polishing rate enhanced with the concentration of the Fe(NO_3_)_3_ due to accelerated surface interactions. However, higher Fe(NO_3_)_3_ concentrations reduce the attractive electrostatic force between the positively charged wet ceria abrasives and the negatively charged deformed surface of the polyimide film, thereby decreasing the mechanically dominant polishing rate. A comprehensive investigation of the chemical and mechanical polishing rate dynamics revealed that the optimal Fe(NO_3_)_3_ concentration to achieve the maximum polyimide film removal rate was 0.05 wt%.

## 1. Introduction

With the advent of the Fourth Industrial Revolution, the rapid growth of markets such as artificial intelligence (AI), metaverse, the internet of things (IOT), and robots has driven increasing demand for semiconductors with a higher operation speed, a lower power consumption, and a higher bandwidth, requiring the rapid scaling-down of advanced memory and logic devices [1,2]. Recently, continuous scaling-down has been slowed down and limited because of the fabrication complexity of devices. As a solution, 2.5D and 3D heterogeneous package technology has been introduced and intensively researched, being essentially necessary to fabricate the redistribution layer (RDL). RDL plays a critical role by rearranging the input/output (I/O) terminals of chips and optimizing their electrical connections with the substrate [3,4,5,6]. To create the RDL, polyimide (PI) has been widely used as an insulation and surface topography planarization layer in RDL structures due to its excellent thermal stability and ductility [7,8,9,10]. Furthermore, in high-performance packaging, multiple-RDL structures have been essential to support a greater number of I/O pins, enhancing the importance of achieving a uniform PI thickness in multiple-RDL structures [11,12,13]. Generally, a polyimide film is spin-coated, and its thickness is approximately several μm, inducing a higher surface topography of several hundred nm. Thus, chemical–mechanical planarization (CMP) of the surface of a polyimide film with a high topography has been essentially utilized to achieve local and global planarization of its surface [8].

Historically, a mechanically dominant CMP method for polyimide films has been the primary method for planarizing the surface of polyimide films’ topography due to the hardness (i.e., 0.33 GPa) of the polyimide film having a strongly covalently bonded structure. However, in 2021, a groundbreaking study introduced a chemically dominant CMP method for polyimide films by using a hydrolysis reaction to deform the surface of the polyimide film into a softer surface structure, thereby improving the polyimide film’s polishing rate [8]. Despite this being a promising solution, this study employed amine-based polishing rate accelerators, which could be limited in their practical application because these polishing rate accelerators are now classified as environmentally regulated substances. Therefore, it is essential to design environmentally friendly polishing rate accelerators capable of converting the surface of polyimide films into a soft structure to improve the CMP polishing rate.

As a new solution for enhancing the polishing rate in CMP of polyimide films, a wet-ceria-abrasive-based polyimide film CMP slurry containing Fe(NO_3_)_3_ was designed. The Fe(NO_3_)_3_ acts as a polishing rate accelerator by breaking the strong covalent bonds between the pyromellitimide and diphenyl ether copolymers on the surface of the polyimide film, thereby deforming this surface into a soft film. Specifically, the Fe(NO_3_)_3_ facilitates a substitution nucleophilic bimolecular reaction (SN2 reaction) that breaks the C=O bonds on the surface of the polyimide film and forms O–C–Fe bonds, followed by ring-opening and a proton transfer reaction, ultimately transforming the surface of the polyimide film into a soft layer composed of strongly positively charged 1,2,4,5-benzenetetracarbonyliron and slightly negatively charged 4,4′-oxydianiline (ODA) [9,14,15,16,17,18,19,20,21,22,23,24]. The modified surface of the polyimide film was polished using 100 nm diameter wet ceria abrasives, achieving a high polishing rate of over 1000 nm/min. 

To elucidate the mechanism behind the significant enhancement in the polyimide film polishing rate achieved by the designed wet ceria abrasive and the Fe(NO_3_)_3_ CMP slurry, both the chemically dominant polishing and mechanically dominant polishing properties of the CMP slurry were investigated. First, the dependency of polyimide film polishing on the Fe(NO_3_)_3_ concentration of the CMP slurry was evaluated. From a chemically dominant polishing perspective, the slurry adsorption degree on the surface of the polyimide film was measured according to the Fe(NO_3_)_3_ concentration in the CMP slurry. Additionally, the deformation of the polished surface of the polyimide film was observed via x-ray photoelectron spectroscopy (XPS), and the results were used to explain the mechanism of surface deformation after CMP.

## 2. Materials and Methods

### 2.1. Materials

A 2000 nm thick polyimide (PI) film (SAMSUNG Inc., Yongin, Republic of Korea) was thermally cured after being coated onto a glass substrate. At this time, the hardness of the polyimide film was 0.33 GPa [8]. To prepare for subsequent polishing, the glass substrate coated with the polyimide film was turned into 4 cm × 4 cm square pieces. The slurry was composed of 1.0 wt% 100 nm wet ceria abrasive (UB materials Inc., Yongin, Republic of Korea) 0 to 0.1 wt% Fe(NO_3_)_3_ (Sigma-Aldrich Inc., St. Louis, MO, USA) as an accelerator, and 0 to 0.2 wt% picolinic acid (Sigma-Aldrich Inc., St. Louis, MO, USA) as a dispersant. At this time, Fe(NO_3_)_3_ was used as the accelerator to increase the polishing rate of the surface of the polyimide film. The slurry’s viscosity was almost independent of the Fe(NO_3_)_3_ concentration, as shown in Appendix A. The presence of CMP debris on the surface of the polyimide film was investigated using an optical microscope.

### 2.2. CMP Conditions

The CMP was conducted using a CMP polisher (POLI-300, G&P Tech. Inc., Busan, Republic of Korea) with a rectangular-grooved CMP pad (SUBA 400, Nitta Haas Inc., Osaka, Japan) [8]. Before the main polishing, pad break-in was performed using brush conditioner for 30 min, and two dummy wafers were polished as part of the preparation. The head pressure of polishing was 4 psi, a head rotation speed of 87 rpm was applied, and the rotation speed of the CMP pad table was 93 rpm. CMP slurry flow rate of 100 mL/min and a polishing time of 60 s were adopted. Then, deionized water (DIW) buffing for 10 s was performed after to remove any remaining debris. 

### 2.3. Characterization

The polyimide film polishing rate was evaluated using a V-VASE ellipsometer (J.A. Woollam Co., Inc., Lincoln, NE, USA). The morphology and diameter of the primary wet ceria abrasives were measured using scanning electron microscopy (SEM) with an S-4800 (Hitachi High-Tech, Tokyo, Japan) using a 15 kV acceleration voltage. The secondary abrasive diameter and zeta potential of the wet ceria abrasives of the CMP slurry were estimated using an ELSZ2+ scattering particle analyzer, Otsuka Electronics (Tokyo, Japan). The pH and conductivity of the CMP slurries were evaluated using a pH meter from Thermo Fischer Scientific Inc., the ORIONSTAR A211 (Waltham, MA, USA). The formation of chemical covalent bonds on the polished surface of the polyimide film after CMP was analyzed using X-ray photoelectron spectroscopy with a K-Alpha+, Thermo Fisher Scientific Inc. (Waltham, MA, USA), at 12 keV and 6 Ma [8]. The adsorption degree of the CMP slurries onto the polished surface of the polyimide film (i.e., the contact angle) was measured via a contact angle meter from GBX Instruments Ltd. (Dublin, Ireland), the DIGIDROP, where 0.01 mL of the DIW or slurries was dropped onto the film surface. 

## 3. Results

### 3.1. The Dependency of the Polyimide Film Polishing Rate on the Fe(NO_3_)_3_ Concentration of the CMP Slurry

The polyimide (PI) film consists of pyromellitimide and diphenyl ether, with strong C–C, C–O, C=O, and C–N bonds [8]. To enhance the polyimide film polishing rate, it is necessary to break these strong covalent bonds on the surface of the polyimide film, thereby deforming the surface into a soft surface of polyimide. This deformation increases the difference in hardness between the wet ceria abrasives and the deformed surface of the polyimide film, which ultimately boosts the polishing rate of the polyimide film. For example, the hardness of the wet ceria abrasive was 7.0 GPa, while the hardness of the polyimide film prior to CMP was ~0.33 GPa [8], resulting in a hardness difference of ~6.67 GPa. By incorporating the Fe(NO_3_)_3_ into the newly designed polyimide film CMP slurry, during CMP, the chemical reaction between the Fe(NO_3_)_3_ and the surface of the polyimide film caused the polished surface of the polyimide film to be deformed into a softer surface with a hardness below 0.33 GPa. Consequently, the difference in hardness between the wet ceria abrasives and the polished surface of the polyimide film exceeds ~6.67 GPa, thereby significantly enhancing the polishing rate. The polyimide film CMP slurry was formulated using 100 nm diameter wet ceria abrasives, along with Fe(NO_3_)_3_ as a polishing accelerator, picolinic acid as a dispersant, and a pH titrant. The dispersant concentration was increased at the same 1:2 ratio when the Fe(NO_3_)_3_ concentration of the CMP slurry was increased.

When the Fe(NO_3_)_3_ concentration of the CMP slurry increased up to 0.05 wt%, the polishing rate of the polyimide film dramatically rose from 725 nm/min to 1465 nm/min, as shown in Figure 1. However, above Fe(NO_3_)_3_ concentrations of 0.05 wt%, the polishing rate gradually decreased from 1465 nm/min to 1298 nm/min. These results indicate two distinct regions depending on the Fe(NO_3_)_3_ concentration: region I, where the polishing rate increases with the Fe(NO_3_)_3_ concentration, and region II, where the polishing rate decreases. Moreover, comparing the polyimide film polishing rates when using other accelerators, such as a hydrolysis accelerator (e.g., ethanolamine) and oxidants (e.g., ammonium persulfate and periodic acid), Fe(NO_3_)_3_ presented the highest polyimide film polishing rate, as shown in Appendix A. Also, the polishing rate of the PI film was almost independent of the frequency of CMP, indicating that the polishing rate of the PI film was not influenced by the by-products generated during CMP, as shown in Appendix A. To understand this dependency, based on a mechanically dominant polishing perspective, the following properties were measured, depending on the Fe(NO_3_)_3_ concentration, in the polyimide film CMP slurry: the secondary abrasive’s diameter, the abrasive’s zeta potential, and the polished polyimide film’s zeta potential. Otherwise, based on a chemically dominant polishing perspective, the following were estimated as a function of the Fe(NO_3_)_3_ concentration of the CMP slurry: the slurry adsorption degree on the surface of the polished polyimide film and the deformation of the polished surface of the polyimide film.

### 3.2. The Properties of the Slurry in Polyimide Film CMP (i.e., the Zeta Potential of the Polished Surface of the Polyimide Film and the Slurry Adsorption Degree) and Mechanically Dominant Polishing Depending on the Fe(NO_3_)_3_ Concentration

In the polyimide film CMP slurry, the wet ceria abrasive’s secondary diameter remained approximately 170 nm when the Fe(NO_3_) _3_ concentration increased from 0 to 0.05 wt%, as presented in Figure 2. However, as the Fe(NO_3_)_3_ concentration exceeded 0.05 wt%, the wet ceria abrasive’s secondary diameter increased from 170 nm to over 200 nm. Furthermore, when the Fe(NO_3_)_3_ concentration of the CMP slurry was enhanced from 0 to 0.1 wt%, the zeta potential of the wet ceria abrasives decreased from 51.7 mV to 39.0 mV. The zeta potential of the slurry without Fe(NO_3_)_3_ was 51.7 mV. However, when Fe(NO_3_)_3_ dissolved in the CMP slurry into Fe^3^^+^ cations and NO_3_^−^ anions, the NO_3_^−^ anions adsorbed onto the surface of the positively charged wet ceria abrasive, increasing the chemical double layer and thereby reducing the zeta potential. Therefore, when the Fe(NO_3_)_3_ concentration in the slurry was increased, the zeta potential of the wet ceria abrasive significantly decreased. The increase in the chemical double layer of the wet ceria abrasive due to the increased Fe(NO_3_)_3_ concentration of the slurry led to an increase in the secondary abrasive diameter. Moreover, when the Fe(NO_3_)_3_ concentration of the slurry was increased from 0 to 0.1 wt%, the zeta potential of the polished surface of the polyimide film noticeably increased from −15.0 mV to −3.3 mV. This increase was attributed to the adsorption of ionized Fe^3+^ from the Fe(NO_3_)_3_ in the CMP slurry onto the negatively charged surface of the polyimide film. As a result, the zeta potential of the polished surface of the polyimide film shifted positively when the Fe(NO_3_)_3_ concentration of the CMP slurry was increased. This was further confirmed through a surface chemical analysis of the polished polyimide film. By analyzing the dependence of the zeta potential of both the wet ceria abrasives and the polished surface of the polyimide film on the Fe(NO_3_)_3_ concentration, the attractive electrostatic force between the wet ceria abrasives and the polished surface of polyimide was calculated to evaluate the mechanically dominant CMP polishing rate during CMP of the polyimide film. When the Fe(NO_3_)_3_ concentration of the slurry was increased from 0 to 0.1 wt%, the attractive electrostatic force between the wet ceria abrasives and the polished surface of the polyimide film decreased from 764 abs. to 126 abs. This result implies that an increase in the Fe(NO_3_)_3_ concentration reduces the attractive electrostatic force between the positively charged wet ceria abrasives and the negatively charged surface of the polyimide film, thereby decreasing the mechanically dominant CMP polishing rate. It is generally known that a higher attractive electrostatic force between abrasives and a polished film surface leads to a higher mechanically dominant polishing rate [24,25,26,27,28,29,30,31,32]. Note that when the Fe(NO_3_)_3_ concentration increased from 0 to 0.1 wt%, the zeta potential of the PI film decreased from −15.0 to −3.3 mV, i.e., a 11.7 mV reduction, while the zeta potential of the abrasives decreased from ~51.7 to 39.0 mV, i.e., a 12.7 mV reduction. Thus, both the zeta potentials of the PI film and abrasives affected the electrostatic attractive force between the PI film surfaces and the abrasives. Thus, from a mechanically dominant polishing perspective, an increase in the Fe(NO_3_)_3_ concentration of the CMP slurry leads to a decrease in the mechanically dominant polishing rate. This result was not correlated with the dependence of the polishing rate on the Fe(NO_3_)_3_ concentration, as observed in Figure 1. Instead, it aligned with the trend observed in region II of Figure 1, where the mechanically dominant polishing rate decreased when the Fe(NO_3_)_3_ concentration was increased. Therefore, to interpret the polyimide film polishing rate depending on the Fe(NO_3_)_3_ concentration, as observed in Figure 1, further observation of the dependence of the chemically dominant polishing rate on the Fe(NO_3_)_3_ concentration is required.

The chemically dominant polishing rate is primarily determined by the adsorption degree of the CMP slurry on the polished surface of the polyimide film (i.e., called contact angle or hydrophilicity) and the chemical reaction at the interface between the CMP slurry chemical and the polished surface of polyimide during CMP (i.e., the deformation of the polished surface of the polyimide film). To measure the slurry adsorption degree on the polished surface of the polyimide film, on changing the Fe(NO_3_)_3_ concentration in the CMP slurry, the contact angle was immediately estimated after dropping 0.01 mL of the slurry onto the polished surface of the polyimide film. The contact angle without Fe(NO_3_)_3_ was 36°, as presented in Figure 3(i). When the Fe(NO_3_)_3_ concentration of the CMP slurry was increased from 0 to 0.10 wt%, the contact angle decreased from 36° to 28°. This result indicates that the hydrophilicity of the polished surface of the polyimide film also increases when the Fe(NO_3_)_3_ concentration of the slurry increases. In general, a decrease in the contact angle of the CMP slurry on the polished film’s surface means an increase in the hydrophilicity. This result suggests that an increase in the Fe(NO_3_)_3_ concentration in the CMP slurry enhances the chemical reaction at the interface between the CMP slurry chemical and the polished surface of the polyimide film. To check this result, the dependency of the CMP slurry’s viscosity on the Fe(NO_3_)_3_ concentration was observed using a viscosity meter. The slurry’s viscosity remained constant despite the increase in the Fe(NO_3_)_3_ concentration, as presented in Appendix A. Therefore, the decrease in the contact angle of the CMP slurry on the surface of the polyimide film with an increasing Fe(NO_3_)_3_ concentration is due to an increase in the chemical reactions at the interface between the slurry and the polished surface of the polyimide film rather than any changes in the slurry’s viscosity.

The dependency of the contact angle on the Fe(NO_3_)_3_ concentration implies that when the Fe(NO_3_)_3_ concentration increases, the hydrophilicity of the polished surface of the polyimide film is also increased, leading to an increase in the chemically dominant polishing rate. Thus, during CMP of the polyimide film, the chemically dominant polishing rate increases with the Fe(NO_3_)_3_ concentration. However, this result was not consistent with the dependency of the polishing rate on the Fe(NO_3_)_3_ concentration shown in Figure 1. Rather, it was related to region I in Figure 1. By simultaneously considering both the dependency of the mechanically dominant polishing rate (i.e., the attractive electrostatic force) and that of the chemically dominant polishing rate (i.e., the contact angle) on the Fe(NO_3_)_3_ concentration, the dependency of the polyimide film polishing rate on the Fe(NO_3_)_3_ concentration in Figure 1 can be interpreted as follows. When the Fe(NO_3_)_3_ concentration of the CMP slurry is enhanced, the mechanically dominant polishing rate is reduced, while the chemically dominant polishing rate is increased. Consequently, the polyimide film polishing rate reaches its maximum with 0.05 wt% of Fe(NO_3_)_3_ in the CMP slurry. In region I of Figure 1, the chemically dominant polishing mechanism is more significant, whereas in region II, the mechanically dominant polishing mechanism plays a more dominant role.

### 3.3. The Surface Deformation of the Polyimide Film Depending on the Fe(NO_3_)_3_ Concentration of the CMP Slurry

To prove that the chemical reaction at the interface between the Fe(NO_3_)_3_ and the surface of the polyimide film (i.e., the deformation of the polished surface of polyimide) increased with the Fe(NO_3_)_3_ concentration of the CMP slurry, the dependency of the chemically reacted bonds in the polished surface of the polyimide film was investigated using XPS. First, the C 1s XPS intensity peak of the polished surface of the polyimide film was analyzed to measure the relative intensities of covalent bonds of C–C, C–O, and O–C–Fe. The binding energies for C–C, C–O, and O–C–Fe bonds were found to be 284.8 eV, 288.52 eV, and 282.46 eV, respectively, as presented in Figure 4a [33,34]. When the Fe(NO_3_)_3_ concentration was increased up to 0.1 wt%, the relative bonding intensity of the C–C bond significantly decreased from 73,658 a.u. to 26,415 a.u., while the relative intensity of the C–O bond slightly increased from 7242 a.u. to 8523 a.u. In addition, the relative bonding intensity of the O–C–Fe bond noticeably increased from 0 a.u. to 5041 a.u., as presented in Figure 4b. In general, the surface of the polyimide film prior to CMP was composed of copolymers of pyromellitimide and diphenyl ether, with C–C, C–O, and C=O covalent bonds. Due to the covalent bonds in the polyimide film’s surface, the chemical reaction at the interface between the surface of the polyimide film and the CMP slurry was limited, resulting in a low polishing rate during CMP. However, the newly designed CMP slurry containing the Fe(NO_3_)_3_ ionized toward Fe^3+^ and NO_3_^−^. During CMP, the Fe^3+^ diffused toward the polished surface of the polyimide film and chemically reacted with the polished surface of the polyimide film, resulting in the formation of O–C–Fe bonds on the polished surface of the polyimide film. Thus, the relative intensities of the O–C–Fe and C–O bonds increased with the Fe(NO_3_)_3_ concentration, while that of the C–C bonds decreased with the increasing Fe(NO_3_)_3_ concentration. Second, the O 1s XPS spectra of the polished surface of the polyimide film were observed to measure the relative intensity of the C–O and C=O bonds. The binding energies for the C–O and C=O bonds were found to be 532.58 eV and 530.2 eV, respectively, as presented in Figure 4c [35,36]. When the Fe(NO_3_)_3_ concentration was increased up to 0.1 wt%, the relative intensity of the C–O bond significantly increased from 63,122 a.u. to 77,539 a.u., while the relative intensity of the C=O bond slightly decreased from 41,593 a.u. to 35,329 a.u., as presented in Figure 4d. Comparing the C 1s XPS spectra of the C–C, C–O, and O–C–Fe bonds in Figure 4b with the O 1s XPS spectra of the C–O and C=O bonds in Figure 4d, it is evident that when the Fe(NO_3_)_3_ concentration was increased, the relative intensities of the C–C bond and the C=O bond evidently decreased, whereas the relative intensities of the C–O bond and the O–C–Fe bond obviously increased. This result indicates that during CMP, the Fe^3+^ in the CMP slurry would undergo chemical reactions with the surface of the polyimide film, i.e., forming O–C–Fe bonds, reducing the C–C bonds, and increasing the C–O bonds via an SN2 reaction, a ring-opening reaction, and a proton transfer reaction [14,15,16,17,18,19,20,21,22,23]. This mechanism will be explained in detail later. Third, the N 1s XPS spectra of the polished surface of the polyimide film were investigated to estimate the relative intensity of the C–NH bond with a binding energy of 399.5 eV, as presented in Figure 4e [37,38]. When the Fe(NO_3_)_3_ concentration was increased up to 0.1 wt%, the relative intensity of the C-NH bond notably increased from 1903 a.u. to 7658 a.u., as presented in Figure 4f. This result indicates that prior to CMP, the surface of the polyimide film contains C–N–C bonds within the pyromellitimide structure. During CMP, the Fe^3^^+^ in the slurry chemically reacts with the surface of the polyimide film, breaking the C–N–C bonds and forming strongly positively charged 1,2,4,5-benzenetetracarbony iron and slightly negatively charged 4,4′-oxydianiline (ODA). The H^+^ in the CMP slurry then reacts with the nitrogen atoms in ODA, forming C–NH bonds on the polished surface of the polyimide film. Thus, the relative intensity of the C–NH bond increases with the Fe(NO_3_)_3_ concentration. Finally, the Fe 2p XPS spectra of the polished surface of the polyimide film were analyzed to calculate the relative intensities of Fe–O @ Fe 2p 3/2, Fe–C and Fe @ Fe 2p 3/2, and Fe–O @ Fe 2p 1/2 bonds. The binding energies were found to be 710.16 eV, 708.45 eV, 716.06 eV, and 724.11 eV, respectively, as presented in Figure 4g [39,40]. These bonds were not observed in the CMP slurry without Fe(NO_3_)_3_, but their relative intensities increased almost linearly when the Fe(NO_3_)_3_ concentration was increased, as presented in Figure 4h. The sequence of a higher XPS peak was followed by Fe–O @ Fe 2p 3/2, Fe–C and Fe @ Fe 2p 3/2, and Fe–O @ Fe 2p 1/2. This result implies that the dissolved Fe^3+^ diffuses to the surface of the polyimide film during CMP and undergoes chemical reactions, forming Fe-O @ Fe 2p 3/2 and Fe–C bonds. Thus, the XPS peak intensities of Fe–O @ Fe 2p 3/2, Fe–C and Fe @ Fe 2p 3/2, and Fe–O @ Fe 2p 1/2 increased with the Fe(NO_3_)_3_ concentration. In summary, when the CMP slurry contains Fe(NO_3_)_3_ in a slightly acidic environment (i.e., at a pH of 5), the Fe(NO_3_)_3_ dissociates into Fe^3+^ and NO_3_^−^, which diffuse and adsorb onto the surface of the polyimide film. During CMP, the Fe^3+^ chemically reacts with the C–C, C–O, C=O, and C–N bonds in the copolymer structure of the pyromellitimide and diphenyl ether on the surface of the polyimide film. These changes lead to the surface deformation of the polyimide film into strongly positively charged 1,2,4,5-benzenetetracarbonyliron and slightly negatively charged ODA, thereby increasing the chemically dominant polishing rate of the surface of the polyimide film. This mechanism will be reviewed in detail later.

### 3.4. The Deformation Mechanism of the Surface of the Polyimide Film During CMP and the Polishing Rate Enhancement via the Effect of Fe(No_3_)_3_

To explain the mechanism according to which the chemical reaction at the interface between the Fe(NO_3_)_3_ and the surface of the polyimide film increased the chemically dominant polishing rate in CMP of the polyimide film, a step-by-step analysis of the reaction process was conducted. First, when Fe(NO_3_)_3_ is added to the CMP slurry, which consists of H_2_O, H^+^, and positively charged (~51.7 mV) wet ceria abrasives at a pH of 5, the Fe(NO_3_)_3_ dissociates into Fe^3+^ and NO_3_^−^, as presented in Figure 5a. Notably, when the Fe(NO_3_)_3_ concentration is increased, the concentration of Fe^3+^ in the slurry also increases. Second, before CMP, the polyimide film is a copolymer composed of pyromellitimide and diphenyl ether, containing strong C–C, C–O, C=O, and C–N covalent bonds, as presented in Figure 5b. During CMP, the positively charged Fe^3+^ diffuses to the negatively charged polished surface of the polyimide film through attractive electrostatic interactions and adsorbs onto the surface. After adsorption, the Fe^3^^+^ encounters the wet ceria abrasives and engages in the rubbing process, leading to polishing, as presented in Figure 5b. Third, the friction energy generated during polishing causes the adsorbed Fe^3^^+^ ions to react with the C=O bonds in the pyromellitimide, which is an SN2 reaction. In this reaction, the Fe^3+^ ion donates an electron to the oxygen atom in the C=O bond, creating a singly negatively charged O–C–Fe^4+^ bond, as presented in Figure 5c. Fourth, the oxygen atom in the O–C–Fe^4+^ bond transfers its non-bonding electrons to the nitrogen atom in the O–C–N–C bond, resulting in the de-bonding of the O–C–N bond and the formation of a ring-opening reaction, creating a singly negatively charged pyromellitimide structure composed of N–C_2_ bonds and O=C–Fe^4+^ bonds. The negatively charged N–C_2_ bond then forms a covalent bond with the diphenyl ether, as presented in Figure 5d. Fifth, the negatively charged N–C_2_ bond in pyromellitimide reacts with the H^+^ ions in the slurry through a proton transfer reaction, forming a H–N–C_2_ bond. One of the carbon atoms in the H–N–C_2_ bond forms a double covalent bond with an oxygen atom, resulting in the H–N–C_2_=O bond. The doubly covalently bonded oxygen atom in the H–N–C_2_=O bond undergoes further reactions through the SN2 reaction, as presented in Figure 5e,f. Sixth, a similar chemical reaction occurs, as presented in Figure 5c,d, leading to the de-bonding of the negatively charged diphenyl ether to the positively charged 4,5-dicarbonyl-iron phthalimide, as presented in Figure 5g. This reaction results in the formation of the negatively charged 4-aminophenyl ether and the positively charged 4,5-dicarbonyl-iron-phthalimide. Since the polyimide film is composed of the copolymers of pyromellitimide and diphenyl ether, the reactions described in Figure 5a–g are repeated in Figure 5h,i. As a result, the polished surface of the polyimide film is converted into slightly negatively charged ODA (4,4′-oxydianiline) and strong positively charged 1,2,4,5-benzenetetracabonyliron with covalent bonds to four Fe^4+^ ions. The 1,2,4,5-benzenetetracarbonyliron exhibits a strong positive charge due to its covalent bonding with four Fe^4^^+^ ions, whereas the ODA remains doubly negatively charged, reducing the zeta potential of the polished surface of the polyimide film. Comparing the film before CMP in Figure 5b with it after CMP in Figure 5j, the C–C, C–O, C=O, and C-N bonds on the surface of the polyimide film prior to CMP were deformed into C–C, C–O, C=O, C–N, and O–C–Fe bonds by CMP. Particularly, comparing between before CMP in Figure 5b and after CMP in Figure 5j, the number of C=O bonds decreased from 4 to 2 after CMP; thus, the relative intensity of the C=O bonds decreased after CMP, and it decreased noticeably with the Fe(NO_3_)_3_ concentration in the CMP slurry. Otherwise, the number of C–O bonds increased from 1 to 3 after CMP; thus, the relative intensity of the C–O bonds increased after CMP, and it increased significantly with the Fe(NO_3_)_3_ concentration. Moreover, O–C–Fe bonds were newly produced after CMP; thus, they increased significantly with the Fe(NO_3_)_3_ concentration in the CMP slurry. Furthermore, the number of C–N–H bonds increased from 0 to 2; thus, the relative intensity of the C–N–H bonds increased after CMP, and it increased remarkably with the Fe(NO_3_)_3_ concentration. All of these changes in the covalent bonds on the surface of the polyimide film after CMP are referred to as polyimide film deformation. As a result, after CMP using the slurry including Fe(NO_3_)_3_, the surface of the polyimide film, consisting of pyromellitimide–diphenyl and diphenyl ether copolymers before CMP, was deformed into slightly negatively charged 4,4′-oxydianiline (ODA) and strong positively charged 1,2,4,5-benzenetetracarbonyliron, with four Fe^4+^ ions.

Before CMP, the surface of the polyimide film had a strong negative zeta potential of −15.0 mV. However, during CMP of the polyimide film, when the Fe(NO_3_)_3_ concentration was increased up to 0.1 wt%, the polished surface of the polyimide film became more positively charged, reducing the zeta potential from −15.0 mV to −10.0 mV, as presented in Figure 2. Moreover, before CMP, the wet ceria abrasive in the CMP slurry had a strong positive zeta potential of +51.7 mV. During CMP of the polyimide film, when the Fe(NO_3_)_3_ concentration in the slurry was increased up to 0.1 wt%, the wet ceria abrasives in the CMP slurry became more negatively charged, reducing the zeta potential from +51.7 mV to +39.0 mV, as presented in Figure 2. Thus, the attractive electrostatic force at the interface between the positively charged wet ceria abrasives and the negatively charged surface of the polyimide film during CMP evidently decreased with an increasing Fe(NO_3_)_3_ concentration. This result showed that the mechanically dominant polishing rate of the polyimide film decreased noticeably with the increasing Fe(NO_3_)_3_ concentration in the CMP slurry. In addition, before CMP, the surface of the polyimide film was composed of strong covalent bonds (i.e., C–C, C–O, C=O, and C–N bonds) in the pyromellitimide–diphenyl ether copolymer, resulting in a low polishing rate of approximately 700 nm/min. However, during CMP, the addition of the Fe(NO_3_)_3_ to the slurry caused the surface of the polyimide film to undergo surface deformation, converting the copolymer structure into slightly negatively charged ODA and strong positively charged 1,2,4,5-benzenetetracarbonyliron. As a result, the hardness of the polished surface of the polyimide film reduced so that the chemically dominant polishing rate increased. Therefore, when the Fe(NO_3_)_3_ concentration in the polyimide film CMP slurry was increased, the chemically dominant polishing rate also increased significantly. However, as a trade-off between the mechanically dominant and chemically dominant polishing rates depending on the Fe(NO_3_)_3_ concentration, the maximum polishing rate of approximately 1500 nm/min occurred at a specific Fe(NO_3_)_3_ concentration of 0.05 wt%, as presented in Figure 1. Note that the amount of passivation of Fe^3+^, i.e., reducing the PI film polishing rate, increased slightly with the concentration of the ferric catalyst in the CMP slurry, as shown in Figure 4. Otherwise, the electrostatic repulsive force, i.e., decreasing the PI film polishing rate, decreased remarkably with the ferric catalyst concentration in the CMP slurry, as shown in Figure 2. Thus, in region II, the decrease in the PI film polishing rate is mainly associated with a decrease in the electrostatic repulsive force rather than an increase in the amount of passivation.

## 4. Conclusions

The continuous scaling-down of memory and logic devices has been essential to improve their operation speeds and reduce their power consumption. However, the rapid increase in the complexity of the fabrication process has slowed down the scaling process, pushing advanced memory and logic devices toward their physical limits. To address these challenges, heterogeneous 2.5D and 3D packaging technology has emerged as a promising solution. In 2.5 and 3D packaging, the redistribution layer (RDL) requires the use of polyimide films with a micrometer-scale thickness. However, spin-coating polyimide films at such thicknesses results in a significant surface topography, necessitating chemical–mechanical planarization (CMP) for effective planarization. Achieving high polishing rates (e.g., >1000 nm/min) is critical for the CMP of polyimide films to meet the industrial requirements. Polyimide films, composed of pyromellitimide and diphenyl ether copolymers, possess strong covalent bonds that contribute to their high surface hardness (i.e., 0.33 GPa). As a result, relying solely on mechanically dominant polishing yields insufficient removal rates, typically below 800 nm/min. Thus, it is essential to design CMP slurry with effective polishing rate accelerators to enhance the polyimide film polishing rate. In this study, Fe(NO_3_)_3_ was introduced into the polyimide film CMP slurry as a polishing rate accelerator. The Fe^3+^ was adsorbed onto the surface of the polyimide film due to the attractive electrostatic force between the positively charged Fe^3+^ ions and the negatively charged polyimide film. During polishing, the wet ceria abrasives induce further surface reactions, including SN2 substitution, ring-opening, and proton transfer reactions, which deform the surface of the polyimide film by converting the pyromellitimide and diphenyl ether groups into strongly positively charged 1,2,4,5-benzenetetracarbonyliron and weakly negatively charged ODA moieties. This deformation reduces the surface hardness of the polyimide film, thereby increasing the chemically dominant polishing rate. However, the NO_3_^−^ ions generated from the dissociation of Fe(NO_3_)_3_ adsorbed onto the positively charged wet ceria abrasives, reducing their positive zeta potential. Additionally, the formation of O–C–Fe^4+^ bonds on the deformed surface of the polyimide film diminishes the negative zeta potential of the polyimide film. As a result, the attractive electrostatic force at the interface between the positively charged wet ceria abrasives and the negatively charged deformed surface of the polyimide film weakens, thereby decreasing the mechanically dominant polishing rate. This study demonstrates that increasing the Fe(NO_3_)_3_ concentration in the CMP slurry enhances the chemically dominant polishing rate while reducing the mechanically dominant polishing rate. Therefore, the overall polishing performance depends on achieving an optimal balance between these two rates. The results show that at a specific Fe(NO_3_)_3_ concentration (i.e., 0.05 wt%), the maximum polyimide film removal rate (i.e., 1465 nm/min) can be achieved. This finding provides a valuable basis for optimizing CMP slurries in advanced RDL fabrication, enabling the efficient planarization of polyimide films in heterogeneous 3D packaging applications.

## Figures and Tables

**Figure 1 nanomaterials-15-00425-f001:**
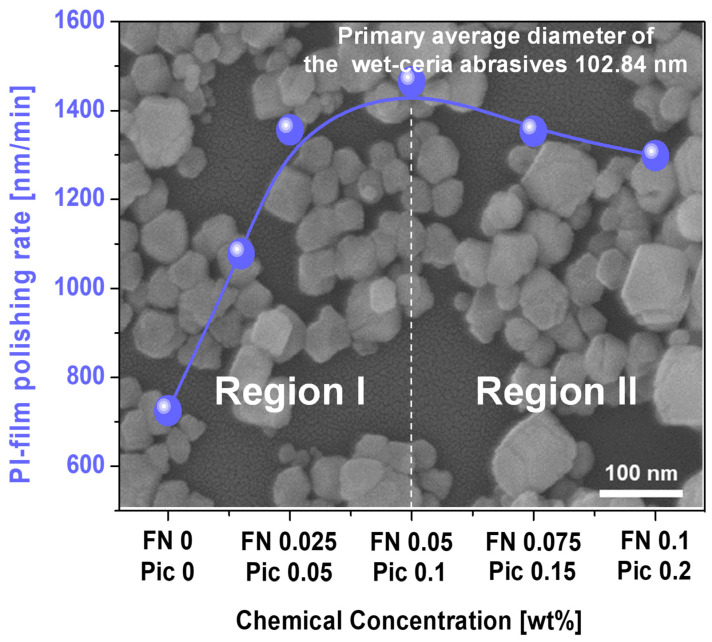
Polyimide film polishing rate depending on concentration of Fe(NO_3_)_3_ and picolinic acid in the CMP slurry. The background SEM images in Figure 1 exhibit the primary abrasive morphology and diameter of the wet ceria abrasives.

**Figure 2 nanomaterials-15-00425-f002:**
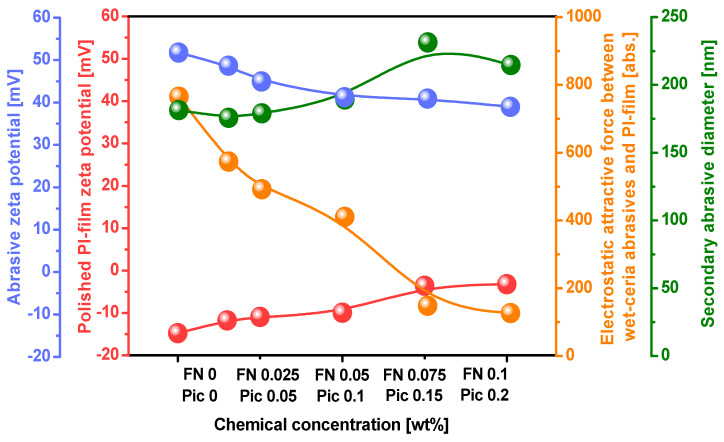
Properties of polyimide film CMP slurry: the secondary abrasive diameter and zeta potentials of the wet ceria abrasives and the polished surface of the polyimide film, varying depending on the Fe(NO_3_)_3_ and dispersant (i.e., picolinic acid) concentrations. The polyimide film CMP slurry was titrated at a pH of 5. The electrostatic attractive force was calculated relative to the zeta potentials of the wet ceria abrasives and the polished surface of the polyimide film.

**Figure 3 nanomaterials-15-00425-f003:**
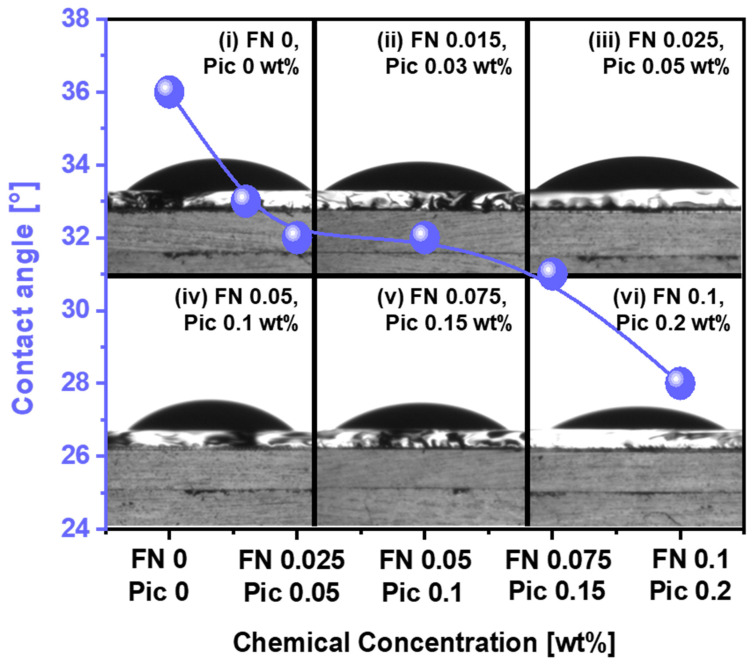
The slurry adsorption degree (i.e., contact angle) depending on the Fe(NO_3_)_3_ concentration in the polyimide film CMP slurry. (i) Slurry without Fe(NO_3_)_3_ and slurries with (ii) 0.015, (iii) 0.025, (iv) 0.05, (v) 0.075, and (vi) 0.1 wt% of Fe(NO_3_)_3_ dropped onto the polished surface of polyimide.

**Figure 4 nanomaterials-15-00425-f004:**
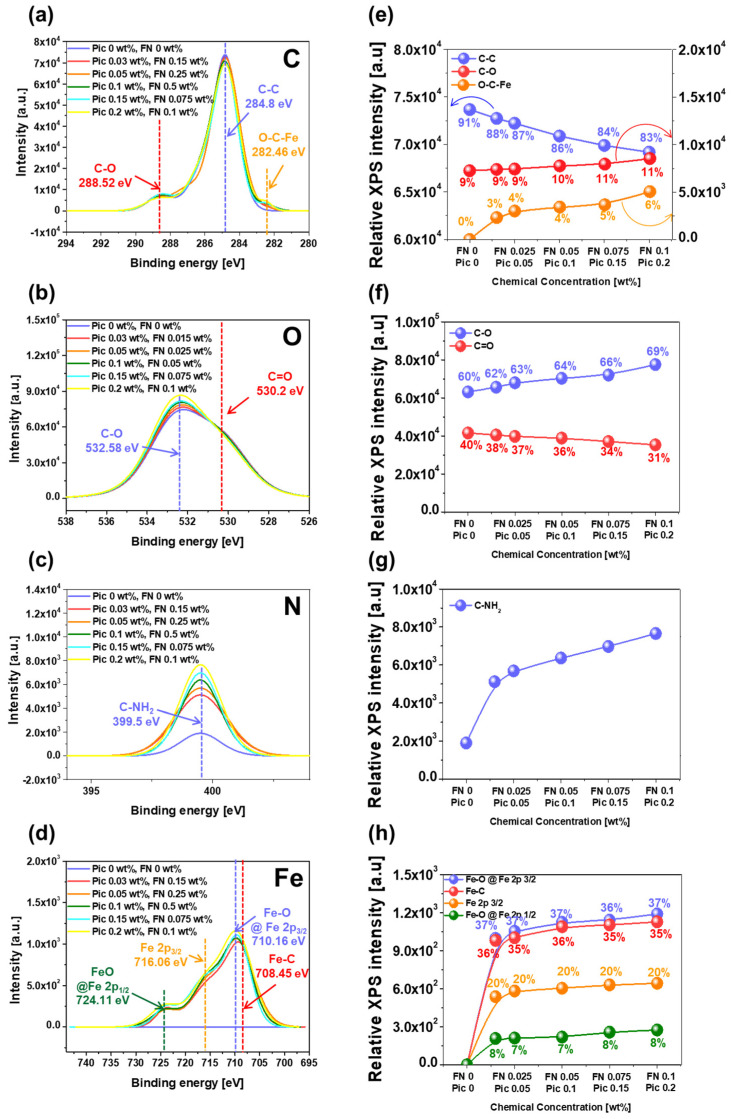
Chemical deformation of the polished surface of polyimide depending on the Fe(NO_3_)_3_ concentration of the slurry. (**a**) C 1s, (**b**) O 1s, (**c**) N 1s, and (**d**) Fe 2p XPS spectra. (**e**) Relative C–C, C–O, and O–C–Fe, (**f**) C–O and C=O, (**g**) C–NH, and (**h**) Fe–O @ Fe 2p 3/2, Fe–C, Fe 2p 3/2, and Fe–O @ Fe 2p 2/1 bond intensities.

**Figure 5 nanomaterials-15-00425-f005:**
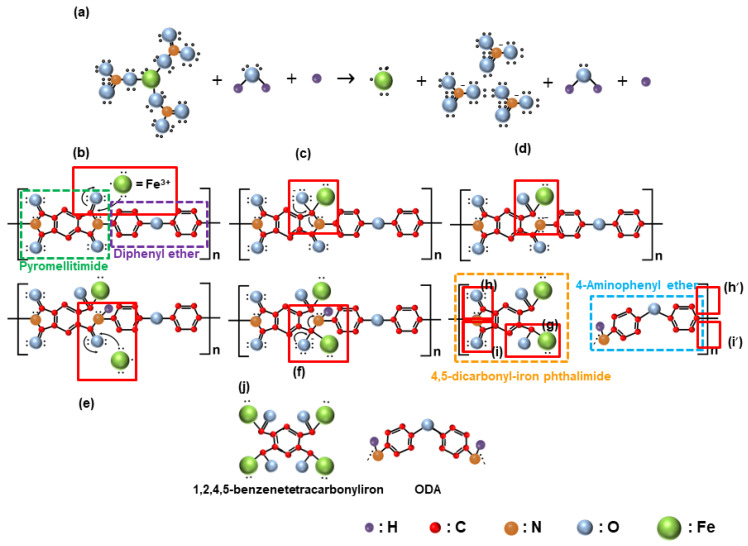
The mechanism of the chemical decomposition of the hard surface of the polyimide film into a soft surface (i.e., positively charged 1,2,4,5-benzenetetracarbonyliron and negatively charged ODA) through CMP using the slurry containing Fe(NO_3_)_3_ (**a**–**j**).

## Data Availability

The raw data supporting the conclusions of this article will be made available by the authors on request.

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
