# Peer review of "The Chemical Deformation of a Thermally Cured Polyimide Film Surface into Neutral 1,2,4,5-Benzentetracarbonyliron and 4,4′-Oxydianiline to Remarkably Enhance the Chemical–Mechanical Planarization Polishing Rate"

_nanomaterials, 2025, doi:10.3390/nano15060425_

Round 1
Reviewer 1 Report
Comments and Suggestions for Authors
The manuscript presents a well-executed study demonstrating how Fe(NO₃)₃ can significantly enhance the CMP rate of polyimide films. The research is innovative and well-supported by experimental data, making it a strong candidate for publication in Nanomaterials.
The novelty of the study should be better contextualized by comparing Fe(NO₃)₃-based CMP with other existing accelerators (e.g., metal ions, oxidative agents, previous hydrolysis approaches).
The mechanistic pathway proposed in Figures 5(a)–5(j) should be cross-referenced with literature on SN2 reactions in polymer degradation for stronger validation.
More discussion is needed on why the rate drops at higher Fe(NO₃)₃ concentrations. Is it due to passivation effects, overloading of Fe³⁺ on the surface, or increased abrasive repulsion?
Does Fe(NO₃)₃ impact slurry viscosity, thereby altering polishing dynamics?
zeta potential of the PI film shifts only slightly, suggesting other electrostatic interactions may be at play. Further discussion is needed to correlate zeta shifts with surface modifications.
Figure 4 presents changes in C, O, N, and Fe bonding states, but the quantification of atomic percentages of these elements before and after CMP is missing.
Does Fe(NO₃)₃ degrade or form unwanted byproducts over multiple CMP cycles?
Reviewer 2 Report
Comments and Suggestions for Authors
This paper is about the chemical and mechanical polishing of PI -ODA copolymer assisted by Fe(NO3)3. The paper is ok, but suffers very much from statements like SN2 and so on (marked in the attachement) plus fugitive and wrong writings as marked in the attachement. All those have to be removed. The extensive explanations about chemical reaction steps fill the paper unnecessarily and should be substituted by chemical drawings, however not using the bulleted structure presented. All explanations on chemistry should be labelled as speculative it they are. It is very confusion to state ion solvatation as decomposition and to state that the reaction as sketched is SN2 without any proof. You are encouraged to correct this

see comments above
Round 2
Reviewer 2 Report
Comments and Suggestions for Authors
Thank you for reworking